# The Role of Home Gardens in Promoting Biodiversity and Food Security

**DOI:** 10.3390/plants12132473

**Published:** 2023-06-28

**Authors:** Helena Korpelainen

**Affiliations:** Department of Agricultural Sciences, Viikki Plant Science Centre, University of Helsinki, P.O. Box 27, FI-00014 Helsinki, Finland; helena.korpelainen@helsinki.fi; Tel.: +358-29-4158383

**Keywords:** biodiversity, home gardens, crop wild relatives, food security, plant genetic resources

## Abstract

Plant genetic resources provide the basis for sustainable agricultural production, adaptation to climate change, and economic development. Many present crop plants are endangered due to extreme environmental conditions induced by climate change or due to the use of a limited selection of plant materials. Changing environmental conditions are a challenge for plant production and food security, emphasizing the urgent need for access to a wider range of plant genetic resources than what are utilized today, for breeding novel crop varieties capable of resilience and adaptation to climate change and other environmental challenges. Besides large-scale agricultural production, it is important to recognize that home gardens have been an integral component of family farming and local food systems for centuries. It is remarkable how home gardens have allowed the adaptation and domestication of plants to extreme or specific ecological conditions, thus contributing to the diversification of cultivated plants. Home gardens can help in reducing hunger and malnutrition and improve food security. In addition, they provide opportunities to broaden the base of cultivated plant materials by harboring underutilized crop plants and crop wild relative species. Crop wild relatives contain a wide range of genetic diversity not available in cultivated crops. Although the importance of home gardens in conserving plant genetic resources is well recognized, there is a risk that local genetic diversity will be lost if traditional plant materials are replaced by high-yielding modern cultivars. This paper provides an overview of home gardens and their present role and future potential in conserving and utilizing plant genetic resources and enhancing food and nutritional security under global challenges.

## 1. Introduction

Plant genetic resources are the basis of sustainable agricultural production, adaptation to climate change and economic development. Besides being critical for food security, they are sources for many other products as well, such as animal feed, fiber, ornamentals, and energy as well as other ecosystem services. Plant genetic diversity contains raw materials for improving the capacity of crops to adapt to climate change and other environmental challenges. It is estimated that, for instance, certain major agricultural crops, e.g., wheat, maize, rice, and soy, may lose up to 25% of their present yield by 2050 due to climate change if varieties adapted to changing conditions are not available by then [1]. Besides decreasing the average yield, climatic drivers, such as drying and warming trends, extreme temperature and precipitation, and carbon dioxide fertilization, are expected to increase yield variability [1]. Despite being crucial for food production, plant biodiversity remains as a relatively little exploited means of breeding crops adapted to new climates [2]. To ensure the continued availability of valuable genetic materials, plant genetic resources can be conserved in wild habitats or on-farm conditions in agroecosystems (both forms of in situ) or outside those (ex situ) in germplasm collections and gene banks. Home gardens are a type of agroecosystem, which are typically small but globally prevalent.

Food production on small plots adjacent to human settlements is the oldest form of cultivation throughout the world [3,4]. The beginning of modern agriculture can be dated back to such subsistence production systems that began in small garden plots around the household as far back as in Early Mesopotamia (10,000 BC) [5]. Still today, these gardens continue to play an important role in local food systems by providing food and income for households [6]. They are an especially prevalent feature of local food systems and the agricultural landscape in developing countries [4]. They provide a continuous supply of fresh vegetables for family use, and they may have an important role in crisis and post-crisis situations [4]. In addition, home gardens have a remarkable role among indigenous communities for food, medicine, and cultural practices, and their deterioration can lead to the erosion of traditional knowledge and practices [4]. Home gardens are not limited to rural settings: in fact, they create the most common form of urban agriculture [7].

Home gardens often consist of multi–layer systems of trees, vegetables, fruits, field crops, spices, herbs, and ornamental and medicinal plants around homesteads [8,9]. Similar to large-scale plant production practices, home gardens are a cropping system composed of soil, crops, weeds, pathogens, and insects, which convert resource inputs, i.e., solar energy, water, nutrients, labor, etc. into food, feed, fuel, fiber, and pharmaceuticals [10]. Although the structure, functions, and contributions of home gardens vary between geographic regions [4], they have some common features, as follows: (1) they are located near a residence, (2) they contain a high diversity of plants, (3) production is supplemental rather than the main source of family consumption and income, (4) they occupy a small area, and (5) the production system is suitable for anybody at some level [6,11].

The vast accessibility of home gardens also provides opportunities for innovative plant production through testing different kinds of plant materials, such as wild materials and local varieties, and through conducting crossing experiments and even fundamental research. The groundbreaking investigations on peas by the monk Gregor Mendel, presented in 1865, were performed in the home garden of a monastery in Brünn and resulted in the formulation of the genetic laws of inheritance that have greatly facilitated the science of genetics and plant breeding [12]. The present paper explores the benefits of home gardens, their importance for food and nutritional security under global challenges, and especially their role and potential in conserving and utilizing genetic resources, as well as enhancing the diversity and adaptation of crop plants.

## 2. Home Gardens Supporting Food and Nutritional Security, and Providing Other Benefits

With an increasing world population, rising urbanization, decreasing arable land, and weather extremes due to climate change, global agriculture is under pressure. Home gardens have an important supporting role when responding to the challenges agricultural production is facing. It is widely recognized that home gardens have multiple functions. They contribute to providing food, proper nutrition, medicine, and other useful products, and they fulfill social and cultural needs, being part of traditional knowledge and practices, while providing different ecosystem services, helping in mitigating climate change effects, and contributing to economic needs and sustainable livelihoods [4,13,14,15,16,17]. It is understandable that attempts to further improve the productivity of these widespread, often eco-friendly, culturally important, and sustainable agricultural practices have been initiated [18,19,20,21].

The most typical function of home gardens is to provide a regular supply of fresh vegetables, which are a very important part of a good diet as they contain various nutrients [22]. Independent production not only saves money but ensures access to a healthy diet that contains adequate macro- and micronutrients. Thus, homestead production of food provides households with direct access to important nutrients that may not be readily available or within their economic reach [4,7]. Cultivation in home gardens has been shown to be associated, for instance, with reduced hunger and malnutrition, improved health, and dietary diversity and balance, including, e.g., an increased consumption of vitamin-A-rich fruits and vegetables, pulses, and other fruits and vegetables [17,23]. Thus, home gardens offer a viable means to improve household food security. It is notable that many uncultivated, as well as neglected and underutilized species and cultivars could make an important contribution to the dietary diversity of local communities [24,25]. In addition, home gardens can be a source of additional income if the household sells a portion of the garden’s produce.

Malnutrition and poor health status are a common problem in developing countries. Specifically, malnutrition among women of reproductive age increases the risk of mortality during pregnancy and puts their newborn children at risk of long-term deficiencies [26]. To overcome this problem, home gardens are considered as a possible solution by contributing to poverty alleviation, improved health, and lower maternal and infant mortality rates [27,28]. In developed countries, home gardening can also decrease the risk of obesity and unhealthy diets [29]. It has been reported that vegetable consumption increased significantly but obesity and metabolic risk decreased as a result of home gardening activities [30]. School gardening projects are an effective way to introduce both plant production and healthier diets [31].

Because of the ongoing global food crisis and rising food prices, there is an increased emphasis on adopting more resilient food systems and strengthening local food production to mitigate the emerging adverse effects [32]. Food production and livelihood enhancement through home gardens can contribute to this effort [7,33]. Such gardens have endured over time as an integral part of local food systems and the agricultural landscape all over the world. Furthermore, home gardens are not limited to rural areas, but they allow the production of fresh, healthy, and inexpensive food in urban settings as well. Whether involving home gardens or other small-scale production systems, urban agriculture can have beneficial environmental and societal impacts, such as a reduction in the urban heat island effect, improved local air quality, improved stormwater management, increased pollinator populations, and climate mitigation services such as carbon sequestration [34].

## 3. The Importance of Plant Genetic Resources

Plant genetic resources are considered to include cultivars, landraces, crop wild relatives (CWR), ecotypes, and genetic stocks (Table 1). They provide the basis for sustainable agricultural production, adaptation to climate change and economic development [35,36,37]. The cultivated crop plants contain only a small part of all plant genetic diversity. An important consequence of crop domestication and bottlenecks created by breeding is that the current gene pool is relatively narrow for most crops, with rather little variation remaining for traits related to resilience and nutritional value. Thus, crop diversification and the development of improved varieties are necessary means for maintaining and stabilizing yields and product qualities, and this is increasingly important under changing climate conditions.

It is recognized that many present crop plants are endangered due to extreme environmental conditions induced by climate change or due to the use of a limited selection of plant materials [38,39]. The increased environmental variability is a challenge for plant production and food security. Without sufficient genetic resources, it will be difficult or even impossible to develop crops that contain important traits, such as pest and disease resistances and the ability to withstand drought, extreme temperatures, and other environmental challenges [2,39,40]. The base of limited genetic resources available in widely used crop plants can be widened by including CWRs, which are taxa related to crops that can potentially donate genetic material with beneficial traits, as well as other underutilized sources. CWRs and wild-harvested plants contain a wide range of genetic diversity and adaptations not available otherwise. Besides being an important part of biodiversity, they carry socioeconomic value and enhance food security [41,42].

In response to the increasing visibility of CWRs in international political agendas since the early 1990’s, a good number of projects have been created and various tools and guidelines have been developed at local, regional, and global levels to enhance their use [43]. It is evident that CWRs are an indispensable asset for breeding programs for expanding and improving the gene pool of cultivated varieties, when novel traits are needed for improvement, e.g., resistance to pathogens and herbivores, nutritional properties, the ability to withstand waterlogging and resilience under changing climatic conditions. Although CWRs have been used for plant breeding for several decades and they have contributed a wide range of beneficial traits, the proportion of CWRs utilized for breeding purposes has, so far, been small. The vast majority of the CWR reservoir remains unexplored [43]. Major obstacles for tapping the genetic diversity of CWRs to improve crops are the hybridization barriers between undomesticated germplasm and the crop, increasing along with divergence, and the extensive and time-consuming pre-breeding work typically required for transferring desirable genetic material to new varieties [44]. On the other hand, recent improvements in the speed of breeding, and high-throughput genotyping and phenotyping make the use of CWR genetic resources more attractive for utilization [41]. Indeed, there are many examples of CWR genes being used to improve crops, such as, wheat, maize, rice, barley, potato, cassava, and legumes [45,46]. Among positive outcomes, the use of CWR genes may lead to a reduced use of pesticides, and sturdier plants which can better manage in competition against weeds, followed by a reduced application of herbicides. Furthermore, improved drought resistance would help saving water by reducing irrigation, and plants with a more efficient use of nutrients need smaller amounts of fertilizers.

An unexpected situation concerning plant genetic resources is their underuse, not overexploitation that threatens their existence [40]. If not being actively used, farmers’ crop varieties as well as those bred by professional plant breeders will not be maintained through continued selection. Rather, they will degrade and may eventually disappear. Yet, such currently underutilized crop plant materials may be able to contribute to climate adaptation and thus are certainly worth further attention. Especially in the case of landraces, home gardens may have an important role in the conservation and utilization of plant genetic resources [15,19,21,24]. The diversity of home gardens is determined by sociocultural (e.g., traditional knowledge and practices) and economic factors, and by climatic and other environmental features [22,47,48]. It is important to learn more of the present role and further potential of home gardens in the management and conservation of a wide range of unique genetic resources for food and on-farm agriculture. So far, many landraces and cultivars, and rare and endangered species have been preserved in home gardens [8,20,49].

Paleoethnobotanical research provides an opportunity to learn about plant use in daily life in an area’s ancient history [50]. It appears that the division between wild and cultivated (or semi-domesticated) food plants was not clearly distinct, since many wild species are thought to have been exposed to various levels of intervention and human management during growth cycles [51]. For instance, Mesoamerican farmers of the village of Joya de Cerén circa 600 CE clearly managed the landscape around their settlements in a manner that would have allowed for harvest from both agricultural and non-agricultural species simultaneously [50]. All weed species recovered from these fields have known uses in nutrition, medicine, or other purposes.

## 4. The Role of Home Gardens in Enhancing the Diversity, Adaptation, and Conservation of Crop Plants

It is remarkable that home gardens have allowed the adaptation and domestication of plants to extreme or specific ecological conditions [24], thus contributing to the diversification of cultivated plants. In such cases, plants develop morphological and physiological characteristics, allowing their adaptation to new or unfavorable habitats. Consequently, home gardens may contain unique and rare locally evolved or developed genetic diversity. An interesting process is introgression, which is used as a breeding tool [52] but which sometimes occurs in home gardens [53]. While such introgression may be beneficial to the productivity of home gardens and perhaps even larger-scale production, it has been clearly proven that gene flow from crop taxa may have a substantial impact on the evolution of wild populations [54]. It has been shown that at least 12 important crops hybridize with wild relatives in some part of their agricultural distribution [54]. These crops include wheat, rice, maize, soybean, barley, cotton, sorghum, millet, beans, rapeseed, sunflower and sugar cane. Crop wild relatives used in cultivation or through introgressive hybridization provide an additional source of plant diversity. Table 2 shows a few examples of evolutionary events observed in local home gardens. It is notable that most such events remain unrecorded for the wider plant breeding community. Further investigations on the composition of home gardens would probably reveal an interesting range of plant materials with adaptations enabling their cultivation in different and changing environmental conditions and use as a material in further breeding, thus promoting resilience and food security.

Landraces developed over time in traditional farming systems, including home gardens, are an underutilized source of genetic variation. One of these species is the common bean (*Phaseolus vulgaris*), domesticated in Mesoamerica and the Andes, but its secondary center of genetic diversity probably extended to Brazil, China, and Europe [55]. After domestication, *P. vulgaris* has become an important crop plant, especially in developing countries. The genetic diversity of Mesoamerican landraces of *P. vulgaris* has been studied, and it has been discovered to possess a very high genetic diversity, which is expected to allow adaptation to diverse environmental conditions, e.g., [56,57]. The proper identification of these novel sources of genetic variation and their use in local breeding efforts can justify and further enhance the conservation of locally adapted beans’ genetic resources in countries where a robust conservation strategy is still missing [56]. In addition, the utilization of wild relatives with specific adaptation traits, such as disease resistances, may be a useful addition to *P. vulgaris* breeding programs. It has been reported in Cuba that a *P. vulgaris* landrace ‘Negrito’ has superior resistance to diseases and harsh weather conditions [53]. Other examples of cultivars discovered in home gardens include several drought-resistant *Allium* landraces from Cuba and salinity-resistant tomato (*Lycopersicon esculentum*) from Guatemala [5].

Most domesticated plant taxa mate with wild relatives somewhere in the world, and gene flow from crop taxa may have a substantial impact on the evolution of wild populations [53,54]. For instance, introgression from the wild tomato *L. esculentum* var. *cerasiforme* to tomato *L. esculentum* has been detected in Cuba [58,59], resulting in interesting variation, including intermediate forms valuable for plant breeding because of fruit characters or disease tolerance. In maize (*Zea mays*), introgressive hybridization and further selection of races have been shown to be common events in the contemporary maize evolution in Cuba [60]. Comparably, introgression among lima bean (*Phaseolus lunatus*) landraces showing intermediate characteristics [61], and introgression between modern varieties and landraces of squash (*Cucurbita moschata*) [60] have been found in Cuban farmers’ fields.

Plant domestication most likely began around human settlements, and the domestication processes of wild plants has continued in home gardens. For instance, the date palm (*Phoenix dactylifera*), which is a widely cultivated species in small-scale home gardens as well as in large plantations and possesses a great number of cultivars, was one of the first fruit trees to be domesticated around 6800–6300 BCE, followed by a complex history of breeding and use [62]. However, such domestication processes have rarely been demonstrated empirically [63]. For instance, for the majority of polyploid crops, it remains uncertain to what extent hybridization and polyploidization preceded domestication or were precipitated by human activities [63]. Some of the crop wild relatives brought into use as a vegetable include the annual green amaranth *Amaranthus viridis* [24], biannual cabbage *Brassica oleracea* [64], and perennial watercress *Nasturtium officinale* [65]. Some herbs, which are considered only marginally important, may have considerable use potential, such as sorrels (genus *Rumex*). Sorrels have been utilized for thousands of years as food, herbal preparations and as a source of different colors of dyes [66]. They are mostly consumed through wild foraging or growing in home gardens. A few types of sorrels are available commercially, including wild types and a few cultivars [66]. There are also wild plants with unusually high tolerance to harsh conditions, e.g., the palm tree *Medemia argun*, which is highly tolerant to drought and heat [67,68]. Its fruit is not considered palatable, and its presently known utilization possibilities are based on the use of its strong fibrous leaves and woody stems for sheltering purposes. However, it is believable that *M. argun* has a wider use potential than presently recognized and it could be a good plant with highly interesting adaptive traits for small-scale home garden production in challenging conditions [67,68].

**Table 2 plants-12-02473-t002:** Reported examples of evolutionary events observed in local home gardens.

**(a) Landraces Adapted to Specific Conditions**				
**Species**	**Landrace**	**Region**	**Reference**	**Comments**
*Allium* sp.	Several landraces	Cuba	Esquivel et al., 1988 [58]	Superior drought resistance
*Lycopersicon esculentum* (tomato)	Several landraces	Guatemala	Esquivel et al., 1988 [58]	Superior salinity resistance
*Phaseolus vulgaris* (common bean)	Negrito	Cuba	Esquivel and Hammer 1992 [53]	Superior resistance to diseases and harsh weather
**(b) Cultivated Plants Developed via Introgression**				
**Taxon 1**	**Taxon 2**	**Region**	**Reference**	**Comments**
*Lycopersicon esculentum*(tomato)	*L. esculentum* var. *cerasiforme* (wild tomato)	Cuba	Esquivel and Hammer 1991 [59]	Fruit characters and disease tolerance
*Phaseolus lunatus* landraces (lima bean)	*P. lunatus* landraces	Cuba	Castiñeiras et al., 1991 [61]	Heterosis in seed characters
*Zea mays* landraces (maize)	*Z. mays* landraces	Cuba	Hatheway 1957 [60]	Hybridization typical in maize evolution
**(c) Crop Wild Relatives Brought into Use**				
**Species**		**Region**	**Reference**	**Comments**
*Amaranthus viridis* (green amaranth)		India	Barbhuiya et al., 2016 [24]	Annual herb/leafy vegetable
*Brassica oleracea* (cabbage)		Romania	Papp et al., 2013 [64]	Biannual plant/diverse vegetable
*Nasturtium officinale* (watercress)		Nepal	Gautam et al., 2006 [65]	Aquatic perennial/leafy vegetable

## 5. How to Improve the Conservation of Biodiversity in Home Gardens?

A good approach to enhance the role of home gardens in promoting biodiversity and food security is to utilize both landraces and wild materials to answer breeding needs created by climate change with higher temperatures and more frequent drought periods. An example is hop (*Humulus lupulus*), which has a long home garden history due to its use in beer production [69]. Future breeding efforts with different quality and adaptation targets are expected to utilize wild hop populations and landraces present in many regions [70]. In fact, good candidates of hop cultivars for growth in warm climates have been found [71]. In addition, it is possible to improve the management practices of genetic resources in home gardens, which would result in a combination of better productivity and superior maintenance of genetic diversity, specifically through the introduction of new crops, improved varieties, and specific characteristics [18]. At the same time, home gardens have a role as important sites of experimentation, plant introduction, and crop improvement as well as refuges for unique genetic diversity [18]. Furthermore, precise molecular characterization is desirable for plant genetic resources, including, to some extent, home garden materials as well. Molecular marker techniques and DNA sequencing allow direct surveys of variation at the DNA level, thereby excluding all environmental influence. Nowadays, these analyses can be performed effectively, even at very early growth stages. Therefore, they have marginalized other methods in genotypic identification [41]. With the onset of DNA and ‘omics’ analyses (i.e., analyses of complete genetic or molecular profiles of organisms based on genomics, transcriptomics, proteomics, or metabolomics), the knowledge of genetic diversity has increased dramatically, as also our understanding of issues, such as domestication, adaptation, and genetic erosion.

Besides the important role of home gardens in conserving plant genetic resources, they contribute to the conservation of biodiversity. These two components are integrated. Especially if plant materials with adequate tolerance and resistance to different biotic stresses are available, the need for the use of pesticides will be reduced, followed by beneficial effects on the biodiversity of the ecosystem in question. This also provides interesting agroecological research possibilities, for instance, the assessment of above- and belowground biodiversity, for which validated and precise tools, such as DNA metabarcoding, are available. It has been shown that the fungal and bacterial biodiversity includes microorganisms, which are potentially beneficial for plant production, e.g., [72,73]. However, the structure and function of fungal and bacterial communities and their interaction and impact on plant performance are still understudied [73].

The ongoing process of evolution in home gardens can be enhanced through selection by farmers, to obtain suitable, adapted plant types to be grown under prevailing and upcoming production conditions. Figure 1 outlines the role of home gardens in conserving plant genetic resources, this being especially important for presently underutilized plant material. Nevertheless, it is important to raise broader awareness of the urgency of conserving plant genetic resources. Although the importance of home gardens in conserving plant genetic resources is well recognized, there is a risk that local genetic diversity will be lost if traditional plant materials are replaced by high-yielding modern cultivars. This type of tendency has been observed, for instance, in Bulgaria, where, however, the local crop diversity of home gardens has remained well preserved until now [21]. Yet, overall, home gardens, when properly managed, have a definite role and importance in the long-term conservation of plant genetic resources.

The conservation of plant genetic resources is based on international agreements, yet each country has a national responsibility to conserve its own genetic resources. One of the goals listed in the draft of the post-2020 Global Biodiversity Framework by the United Nations Environment Programme is that “genetic diversity of wild and domesticated species is safeguarded, with at least 90 per cent of genetic diversity within all species maintained” [74]. Achieving this challenging target will require well-coordinated conservation action at national, regional, and international levels. For decades, gene banks have played a key role in conservation measures when attempting (a) to sustainably conserve the broadest range of genetic diversity found in the target species maintained as population samples or accessions, (b) to characterize and evaluate this diversity to aid selection for utilization, and (c) to make the accessions available to the users [75]. However, it is evident that complementary in situ conservation action is needed. In fact, a systematic application of in situ conservation measures is estimated to at least double the diversity available to users [75].

Home gardens are typically small and privately operated. Despite that, they can contribute to in situ conservation through on-farm conservation measures. Their impact on the conservation of plant genetic resources can be enhanced through better structured and coordinated management measures, in combination with ex situ conservation. In addition, it is important to recognize that home gardens and their plant genetic resources are dynamic systems. Firstly, they are influenced by cultural, social, and economic factors, as home gardens are typically managed by persons having different goals and preferences, which change over time. Secondly, biological evolution functions in on-farm production systems, resulting in changes in the composition of genetic resources over time, possibly generating new adaptations. Such changes are especially important under ongoing climate change. Therefore, besides increasing conservation through using the existing genetic diversity, home gardens and other on-farm systems have an important complementary role in relation to the static ex situ conservation through dynamic evolutionary processes. This is especially important for underutilized crops, which are often neglected in plant breeding activities.

In summation, measures enhancing the conservation of plant diversity present in home gardens include (1) systematic monitoring and documentation of the diversity, dynamic changes in diversity due to ongoing evolution, threats, and the conservation status of plant materials, (2) improving knowledge on valuable traits and genetic characteristics, (3) expanding gene bank coverage of home garden materials, and (4) increasing the availability of these plant materials to both formal and on-farm crop improvement programs.

## Figures and Tables

**Figure 1 plants-12-02473-f001:**
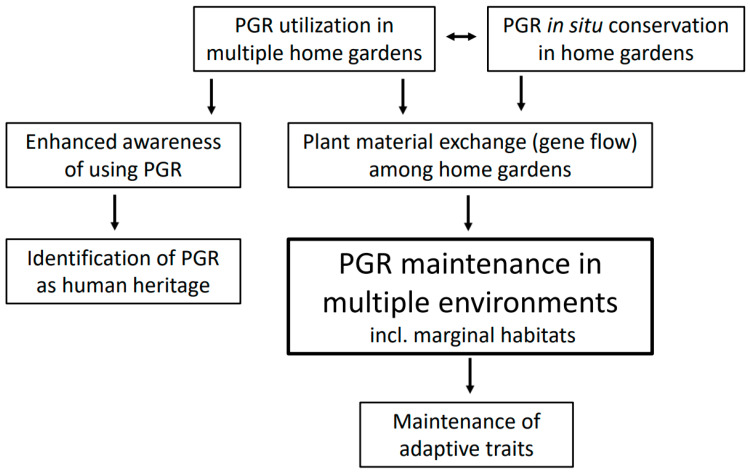
The role of home gardens in conserving plant genetic resources (PGR) in multiple environments through PGR utilization, material exchange and identification as cultural heritage.

**Table 1 plants-12-02473-t001:** Types of plant genetic resources.

Type	Description
Cultivars	Varieties produced by plant breeders, usually uniform and adapted to high farm management standards
Landraces	Varieties developed over time in traditional farming systems, usually variable and adapted to local conditions
Crop wild relatives	Wild taxa within the same genus as a crop
Ecotypes	Populations of wild forms of domesticated species or their wild relative species, or other wild material; specific adaptations
Genetic stocks	Material generally used by research or breeding programs resulting in specific information on a gene or character, or other data of value for breeding and research

## Data Availability

Not applicable.

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
