# Peer review of "The Role of Home Gardens in Promoting Biodiversity and Food Security"

_plants, 2023, doi:10.3390/plants12132473_

Round 1

Reviewer 1 Report

This paper contains a review of the value of home gardens in promoting and conserving broad biodiversity, and the impact they have for food and nutritional security principally in developing countries. The paper is well written with minor edits required to improve the manuscript. For specific edits, please see attachment - I have included only the pages that require edits.

English is very good with only minor edits required.

Author Response

-Thank you for the comments. I have made all edits as suggested.

Reviewer 2 Report

I carefully read the Review “The role of home gardens in promoting biodiversity and food

Security” submitted by  Helena Korpelainen for the special issue “A Critical Review of the Current Approaches and Procedures of Plant Genetic Resources Conservation and Facilitating Use: Theory and Practice.

Although the proposed argument is in line with the special issue, the analysis and evaluation of the work of other experts in the considered field is very low.

Reviews require a high level of detail and a well-structured presentation of the topics. A review is a critical and constructive evaluation of the literature in a particular field through synthesis, classification, analysis and comparison. Authors normally need to use database searches to portray the research and the first goal is to summarize everything and present a clear understanding of the topic the authors have been working on. 

Author Response

-I have performed an extensive search of related literature when writing the paper. I believe the paper is a review within the covered area (not everything possible about plant genetic resources). The attempt was to review the benefits of home gardens, their importance for food and nutritional security, and especially their role and potential in conserving and utilizing genetic resources, as well as enhancing the diversity and adaptation of crop plants. Yet, I have tried to improve the presentation and content. Especially the two new paragraphs at the end of the manuscript provide further conclusions and suggestions.

Reviewer 3 Report

The topic of the paper is very interesting and topical, but it has many gaps.

There is absolutely no reference to international documents protecting landraces in cultivated plants. At the level of the European Union there are numerous regulations related to the same subject.

The paper does not give us an overview of the evolution and current situation of the genetic diversity of crop plants.

Gene banks and seed banks are very important for the conservation of landraces. However, the author does not attach any importance to these preservation methods.

The author avoids presenting figures and percentages related to anything. He only makes pleas not backed by concrete data. Chapter 4 is an exception, however, and here only examples are given, unable to provide a global picture of the issue.

The author uses the catchphrase - agriculture "in situ" - and does not mention anything about agriculture "on farm". I specify that, worldwide and especially European, the second phrase is preferred: "on farm".

L 35-36 In the paper it is stated that "It is estimated that certain major agricultural crops may lose up to 25% of their present yield by 2050 due to climate change if better adapted varieties are then not available..."

What are those crops and why are they more affected than others? It must be explained in the paper. Reference to the bibliographic source is not enough.

L 56-58. The paper states that "Further more, home gardens are specified as a cropping system composed of soil, crops, weeds, pathogens, and insects, which convert resource inputs, i.e., solar energy, water, nutrients, labor, etc. into food, feed, fuel, fiber, and pharmaceuticals". Not only family gardens transform solar energy and other energy inputs into food, biofuels, etc. This also applies to industrial crops. The author needs to clarify what he meant.

L 177-178. The paper states that "Especially in the case of landraces, crop wild relatives and ecotypes, home gardens may have an important role in the conservation and utilization of plant genetic resources...". I don't know if family gardens have an important role in the conservation of wild varieties. As a rule, gardeners prefer to grow varieties or landraces with superior taste qualities. The author needs to clarify what he meant.

L 206-208 In the paper it is stated that "It has been shown that 12 crops out of 13 most important food crops..." What are those cultures?

L 304-305. The paper states that "It has been shown that the fungal and bacterial biodiversity includes microorganisms, which are potentially beneficial for plant production". What kind of fungi and bacteria are they referring to? Even if the author refers to the bibliographic source, he must clearly specify what it is about.

Author Response

The topic of the paper is very interesting and topical, but it has many gaps. There is absolutely no reference to international documents protecting landraces in cultivated plants. At the level of the European Union there are numerous regulations related to the same subject. The paper does not give us an overview of the evolution and current situation of the genetic diversity of crop plants.

--These were not the scope of the paper. However, I have now added related content.

Gene banks and seed banks are very important for the conservation of landraces. However, the author does not attach any importance to these preservation methods.

--I agree about the importance of gene/seed banks, but the paper was focused on home gardens. Yet, I have included additional content.

The author avoids presenting figures and percentages related to anything. He only makes pleas not backed by concrete data. Chapter 4 is an exception, however, and here only examples are given, unable to provide a global picture of the issue.

--I cannot answer anything to this.

The author uses the catchphrase - agriculture "in situ" - and does not mention anything about agriculture "on farm". I specify that, worldwide and especially European, the second phrase is preferred: "on farm".

--I agree that “on farm” is a good phrase and I use it now. Originally, I had chosen “in situ” on purpose to be consistent.

L 35-36 In the paper it is stated that "It is estimated that certain major agricultural crops may lose up to 25% of their present yield by 2050 due to climate change if better adapted varieties are then not available..." What are those crops and why are they more affected than others? It must be explained in the paper. Reference to the bibliographic source is not enough.

--Not only the major crops wheat, maize, rice, and soy are affected, but their yield loss will have a bigger impact on food security. The text is revised.

L 56-58. The paper states that "Further more, home gardens are specified as a cropping system composed of soil, crops, weeds, pathogens, and insects, which convert resource inputs, i.e., solar energy, water, nutrients, labor, etc. into food, feed, fuel, fiber, and pharmaceuticals". Not only family gardens transform solar energy and other energy inputs into food, biofuels, etc. This also applies to industrial crops. The author needs to clarify what he meant.

--Surely true. The text is clarified.

L 177-178. The paper states that "Especially in the case of landraces, crop wild relatives and ecotypes, home gardens may have an important role in the conservation and utilization of plant genetic resources...". I don't know if family gardens have an important role in the conservation of wild varieties. As a rule, gardeners prefer to grow varieties or landraces with superior taste qualities. The author needs to clarify what he meant.

--The text is revised accordingly.

Round 2

Reviewer 2 Report

While I appreciate the author's effort to improve the text,

I find that the manuscript produced is scientifically not valid for Plants journal

My observations remain the same as the first review.

It is a descriptive work that contains little data being a review

Author Response

Thank you for the comment. I still believe that the paper is a review within the covered area (not everything possible about plant genetic resources). The attempt was to review the benefits of home gardens, their importance for food and nutritional security, and especially their role and potential in conserving and utilizing genetic resources, as well as enhancing the diversity and adaptation of crop plants. My second revision includes some further improvements, primarily on the strategic importance of home gardens (marked in yellow in the manuscript).  

Reviewer 3 Report

From my point of view, the paper is fine now. Congratulations!

Author Response

Thank you!